# Topic Models with Survival Supervision: Archetypal Analysis and Neural Approaches

## Abstract

We introduce two approaches to topic modeling supervised by survival analysis. Both approaches predict time-to-event outcomes while simultaneously learning topics over features that help prediction. The high-level idea is to represent each data point as a distribution over topics using some underlying topic model. Then each data point's distribution over topics is fed as input to a survival model. The topic and survival models are jointly learned. The two approaches we propose differ in the generality of topic models they can learn. The first approach finds topics via archetypal analysis, a nonnegative matrix factorization method that optimizes over a wide class of topic models encompassing latent Dirichlet allocation (LDA), correlated topic models, and topic models based on the "anchor word" assumption; the resulting survival-supervised variant solves an alternating minimization problem. Our second approach builds on recent work that approximates LDA in a neural net framework. We add a survival loss layer to this neural net to form an approximation to survival-supervised LDA. Both of our approaches can be combined with a variety of survival models. We demonstrate our approach on two survival datasets, showing that survival-supervised topic models can achieve competitive time-to-event prediction accuracy while outputting clinically interpretable topics.

## 1 Introduction

Predicting time-to-event outcomes arises in a variety of applications. For example, in healthcare, we may be interested in predicting how much time a patient has to live. In criminology, we may be interested in predicting when a convicted criminal might reoffend. In e-commerce and on streaming platforms, companies with subscription services like Amazon and Netflix may be interested in predicting when users might cancel their subscriptions. In many such applications, we can now collect an enormous number of measurements per person/subject. However, how all of these measurements relate is typically unknown. In this paper, we aim to address the twin objectives of learning how measurements relate in the form of a topic model, and learning how topics can assist in predicting time-to-event outcomes via a survival analysis model.

For ease of exposition, we phrase the problem we consider in the classical survival analysis context of predicting time until death. We assume that we have access to a training dataset of $n$ subjects. For each subject, we know how many times each of $d$ "words" appears, where the dictionary of words is pre-specified. As an example, in a clinical context, one word might correspond to "low blood pressure reading"; for a given subject, we can count how many such readings the subject has had recorded in the past. We denote $X_{i,u}$ to be the number of times word $u \in \{1, \ldots, d\}$ appears for subject $i \in \{1, \ldots, n\}$. Viewing $X$ as an $n$-by-$d$ matrix, the $i$-th row of $X$ can be thought of as the feature vector for the $i$-th subject. As for the training label for the $i$-th subject, we have two recordings: event indicator $\delta_i \in \{0, 1\}$ specifies whether the $i$-th subject died, and observed time $Y_i \in \mathbb{R}_+$ is the $i$-th subject's "survival time" (time until death) if $\delta_i = 1$ or the "censoring time" if $\delta_i = 0$. The censoring time gives a lower bound on the survival time for the $i$-th subject. For example, when we stop collecting data, some subjects will still be alive, so we know they live at least as long as when we stopped collecting training data.

Our goal is to discover topics for the $d$ words that help predict survival times of unseen test subjects. Note that an unsupervised topic model like latent Dirichlet allocation (LDA) (Blei et al., 2013) would not use any of the training labels (the event indicators $\delta_i$'s and observed times $Y_i$'s), learning

topics using only the word counts matrix $X$. Meanwhile, in survival analysis, a standard approach would involve learning a survival model using all the patients' feature vectors and labels but the model would not learn thematic structure in the different features, *e.g.*, topics. Jointly learning both a topic model and a survival model was first done by Dawson & Kendziorski (2012), who combined LDA with a Cox proportional hazards model (Cox, 1972). Using LDA with $r$ topics, Dawson and Kendziorski represent the $i$-th subject as a probability vector $W_i \in [0,1]^r$ specifying the subject's membership in each of the $r$ topics; then $W_i$'s are treated as the input covariates to the Cox model. Dawson and Kendziorski called this joint model SURVLDA and derived a variational EM algorithm to estimate its parameters.

In this paper, we build on SURVLDA by proposing two new survival-supervised topic modeling approaches, both of which allow for either the topic or the survival model to be replaced. Our contributions are as follows:

- (Section 3) We show how to take a discriminative approach to jointly learning topic and survival models, where topics are estimated via archetypal analysis (Cutler & Breiman, 1994; Javadi & Montanari, 2019). Archetypal analysis represents each subject as a convex combination of "archetypes", which are optimized to be diverse yet still be close to the convex hull of the subjects' feature vectors. Applied to topic modeling, the archetypes are the topics, with each archetype specifying a particular topic's word distribution. Archetypal analysis does not assume a parametric model and can learn a wide class of topic models. We describe how to combine archetypal analysis with any survival analysis model for which we can take a specific partial derivative.

- (Section 4) We approximate Dawson and Kendziorski's SURVLDA model in a neural net framework, which allows for different choices of topic and survival models to be combined. This approach requires that the topic and survival models already have neural net approximations or formulations. For example, LDA and some variants of it can already be approximated using variational autoencoders (Srivastava & Sutton, 2017; Card et al., 2018). In particular, Card et al. (2018) show how to approximate supervised LDA (McAuliffe & Blei, 2008) in a neural net framework that they call SCHOLAR; they specifically consider classification as the supervised task although they mention that their framework could be used to predict other real-valued outputs. We specifically combine their approach with that of Katzman et al. (2018) to handle survival supervision.

- (Section 5) We apply our two proposed approaches to two survival analysis datasets (predicting how long pancreatitis patients stay in an intensive care unit, and time until death for breast cancer subjects), comparing against a number of classical and recently developed deep survival analysis baselines. Survival-supervised topic models have time-to-event prediction accuracy that is competitive with top-performing existing baselines while producing clinically interpretable topics.

## 2 BACKGROUND

We begin with some background on archetypal analysis, topic modeling, and survival analysis. Along the way, we introduce notation that recurs throughout the paper. As a reminder, we assume that we have access to training data $(X_1, Y_1, \delta_1), (X_2, Y_2, \delta_2), \ldots, (X_n, Y_n, \delta_n)$, where the $i$-th training subject has feature vector $X_i \in \mathbb{R}^d$, observed time $Y_i \in \mathbb{R}_+$, and event indicator $\delta_i \in \{0, 1\}$. Throughout this paper, we generally take $X_{i,u}$ (for $u \in \{1, 2, \ldots, d\}$) to be the number of times word $d$ appears, for some user-specified dictionary of $d$ words. We let $\overline{X}_{i,u}$ denote the fraction of times a word appears for a specific subject, meaning that

$$\overline{X}_{i,u} = \frac{X_{i,u}}{\sum_{v=1}^{d} X_{i,v}}.$$

Note that $\overline{X}$ is an $n$-by-$d$ matrix, and we use $\overline{X}_i$ to denote the $i$-th row of $\overline{X}$. We use this indexing notation for other matrices as well.

### 2.1 ARCHETYPAL ANALYSIS AND TOPIC MODELING

Archetypal analysis (Cutler & Breiman, 1994; Javadi & Montanari, 2019) posits that each training vector $\overline{X}_i$ can be well-approximated by a convex combination of $r$ different unknown "archetypes"

$H_1, H_2, \ldots, H_r \in \mathbb{R}^d$:

$$\overline{X}_i \approx \sum_{g=1}^{r} W_{i,g} H_g \qquad (2.1)$$

for some weights $W_{i,1}, \ldots, W_{i,r} \in [0,1]$ that sum to 1, i.e., the vector $W_i := (W_{i,1}, \ldots, W_{i,r})$ resides in the probability simplex $\Delta^r := \{w \in [0,1]^r : \sum_{g=1}^{r} w_g = 1\}$. By stacking the archetypes $H_1, \ldots, H_r$ as rows to form the matrix $H$, equation (2.1) can be expressed as $\overline{X} \approx WH$. Archetypal analysis aims to estimate $W$ and $H$ given $\overline{X}$.

If the archetypes $H_1, \ldots, H_r$ are constrained to be in the probability simplex $\Delta^d$, then we get a topic model, and each archetype corresponds to a word distribution. For example, if rows of $W$ are generated i.i.d. from a Dirichlet distribution, and rows of $H$ are generated i.i.d. from another Dirichlet distribution, then we get LDA (Blei et al., 2013). As a slight modification of this setup, if the rows of $W$ are instead generated from a logistic normal distribution that allows correlation between topics, we get the correlated topic model (Lafferty & Blei, 2006). For an example that is not generative, if the archetypes are on a probability simplex, and for each archetype $g \in \{1, \ldots, r\}$, there exists a word $w$ that only appears in archetype $g$ (i.e., $H_{g,w} > 0$ and $H_{h,w} = 0$ for all $h \neq g$), then we have a topic model satisfying the separability or "anchor word" assumption (Donoho & Stodden, 2004; Arora et al., 2012a;b; 2013). Archetypal analysis optimizes over matrices $W$ and $H$ that include all of the aforementioned topic models above as special cases. In fact, archetypal analysis does not require that archetypes be on a probability simplex or that they be nonnegative; the input matrix $\overline{X}$ need not consist of word frequencies and could be positive or negative real-valued measurements. Crucially, the error in approximation (2.1) should be small; precise details including identifiability and degeneracy issues can be found in Section 3 of Javadi & Montanari (2019).

To estimate weights $W$ and archetypes $H$, Javadi and Montanari proposed the following approach. First, for a point $u \in \mathbb{R}^d$ and a matrix $V \in \mathbb{R}^{m \times d}$, we define the distance from $u$ to the convex hull of the rows of $V$ as

$$D(u, V) := \min_{w \in \Delta^m} \|u - V^\top w\|_2.$$

The vector $w \in \Delta^m$ that achieves the minimum consists of the convex combination weights that best combine rows of $V$ to approximate the point $u$. Then Javadi and Montanari (approximately) minimize the nonconvex loss

$$L_{\mathsf{arch}}(W, H; \lambda) := \underbrace{\sum_{i=1}^{n} \|\overline{X}_i - H^\top W_i\|_2^2}_{\spadesuit} + \lambda \underbrace{\sum_{g=1}^{r} D^2(H_g, \overline{X})}_{\heartsuit} \qquad (2.2)$$

subject to the constraint that $W_i \in \Delta^r$ for $i = 1, \ldots, n$; constant $\lambda \geq 0$ is a user-specified regularization parameter. Minimizing term $\spadesuit$ (error of approximating input data $\overline{X}_i$'s as convex combination of archetypes) encourages the archetypes to be far apart and have a convex hull that contains the input data. However, this term does not prevent the archetypes from taking on extreme values; for example, if the archetypes already have a convex hull that contains the $\overline{X}_i$'s (so that $\spadesuit = 0$), we can move the archetypes even farther apart and still have their convex hull contain the $\overline{X}_i$'s (so we still have $\spadesuit = 0$). We prevent this behavior by minimizing term $\heartsuit$, which encourages each archetype to be close to the convex hull of the input data.

To learn a topic model, we enforce that the archetypes correspond to distributions over words by requiring each row of $H$ to be in probability simplex $\Delta^d$. The resulting optimization problem is

$$(\widehat{W}, \widehat{H}) \in \operatorname*{argmin}_{\substack{W \in \mathbb{R}^{n \times r},\ H \in \mathbb{R}^{r \times d} \\ \text{s.t. } W_i \in \Delta^r \text{ for all } i,\ H_g \in \Delta^d \text{ for all } g}} L_{\mathsf{arch}}(W, H; \lambda). \qquad (2.3)$$

A local minimum can be found by alternating between minimizing $W$ with $H$ fixed, and vice versa.

## 2.2 SURVIVAL ANALYSIS

Archetypal analysis and topic models are unsupervised methods. To predict time-to-event outcomes, we turn toward survival analysis models. Suppose we take the $i$-th subject's feature vector to be $W_i \in \mathbb{R}^r$ instead of $X_i$. As this notation suggests, when we combine topic and survival models,

$W_i$ corresponds to the $i$-th subject's archetype/topic combination weights; this strategy for combining topic and survival models was also done by Dawson & Kendziorski (2012), which in turn is based on the supervised LDA formulation (McAuliffe & Blei, 2008). We treat the training data as $(W_1, Y_1, \delta_1), \ldots, (W_n, Y_n, \delta_n)$, disregarding the $X_i$ and $\overline{X}_i$ values from earlier.

We aim to reason about the survival time of an unseen test feature vector $W_0 \in \mathbb{R}^r$. Specifically, let random variable $T_0$ denote the survival time corresponding to feature vector $W_0$ (treated as a random variable). Then our goal is to produce an estimate $\widehat{S}$ of the (conditional) survival function

$$S(t|w) := \mathbb{P}(T_0 > t \mid W_0 = w) \qquad \text{for } t \geq 0 \text{ and } w \in \mathbb{R}^r.$$

Importantly, for a given feature vector $w \in \mathbb{R}^d$, note that $S(\cdot|w)$ is a function. If we have an estimate $\widehat{S}(\cdot|w)$ for $S(\cdot|w)$, we can compute a single number for the predicted survival time for feature vector $w$. The basic idea is to find a time $t$ such that $\widehat{S}(t|w) \approx 1/2$; such a time corresponds to a median survival time. Details for computing this median survival time estimate is in Appendix A.

Different survival models place different assumptions on $S$, where we typically assume that the training and test data points are i.i.d. samples from the same underlying distribution. The technical challenge is that in general, we do not see the survival times for all of the training subjects: the observed times $Y_i$'s are equal to survival times only for subjects who have $\delta_i = 1$; all other $Y_i$ values are censoring times. Different censoring models are used. A standard approach is to assume that the $i$-th training subject has survival time $T_i$ and censoring time $C_i$ that are conditionally independent given feature vector $W_i$, and if the survival time occurs before censoring ($T_i \leq C_i$), then $Y_i = T_i$ and $\delta_i = 1$; otherwise $Y_i = C_i$ and $\delta_i = 0$. This setup is referred to as *random censoring*. Details can be found in a survival analysis textbook (e.g., Kalbfleisch & Prentice, 2002).

As a concrete example of how survival function $S$ can be computed via minimizing a loss function, we next present the classical Cox proportional hazards model (Cox, 1972). As we discuss shortly, this is just one example of a survival model that can be combined with our proposed survival-supervised archetyptal analysis or neural topic modeling approaches.

**Example 1 (Cox proportional hazards)** *Recall that survival function $S$ is 1 minus the CDF of the distribution of survival time $T_0$ given feature vector $W_0 = w$. We denote the CDF of this distribution as $F(t|w)$ and assume it has a probability density function $f(t|w) = \frac{\partial}{\partial t} F(t|w)$. Then the Cox model constrains $S$ through the so-called hazard function $h$ of $S$, given by*

$$h(t|w) := -\frac{\partial}{\partial t} \log S(t|w) = \frac{-\frac{\partial}{\partial t}[1 - F(t|w)]}{S(t|w)} = \frac{f(t|w)}{S(t|w)}, \tag{2.4}$$

*which is the instantaneous rate of death at time $t$ divided by the probability of surviving up to time $t$, all conditioned on the feature vector being $w$. Specifically, the Cox model assumes that hazard function $h$ factors as*

$$h(t|w) = h_0(t) e^{\beta^\top w},$$

*where the two parameters are the baseline hazard function $h_0 : \mathbb{R}_+ \to \mathbb{R}_+$, and the vector of regression coefficients $\beta \in \mathbb{R}^r$. Under random censoring (and actually more general censoring models), we can estimate $\beta$ without knowing $h_0$ via maximizing a profile likelihood, which is equivalent to minimizing the loss*

$$L_{\text{Cox}}(\beta|W) := \sum_{i=1}^n \delta_i \Big[ -\beta^\top W_i + \log \sum_{\substack{j=1 \ s.t. \ Y_j \geq Y_i}}^n \exp(\beta^\top W_j) \Big]. \tag{2.5}$$

*Given an estimate of $\beta$, we can deterministically compute a nonparametric estimate for baseline hazard function $h_0$; this estimation procedure is standard and can be found in Section 7.8 of Cox & Oakes (1984). Once we have estimates $\widehat{h}_0$ and $\widehat{\beta}$ for $h_0$ and $\beta$, then for any test feature vector $w \in \mathbb{R}^r$, we can estimate this test feature vector's corresponding hazard function via*

$$\widehat{h}(t|w) = \widehat{h}_0(t) e^{\widehat{\beta}^\top w}.$$

*Using the first equality of equation (2.4), note that $S(t|w) = \exp\big( - \int_0^t h(\tau|w) d\tau \big)$. We can plug estimate $\widehat{h}$ for $h$ into this equation to get an estimate of $S$:*

$$\widehat{S}(t|w) = \exp\Big( - \int_0^t \widehat{h}(\tau|w) d\tau \Big). \tag{2.6}$$

*In practice, the integral is computed via a summation.*

Other survival losses are possible aside from $L_{\mathsf{Cox}}(\beta|W)$. As a second example, we provide the survival loss function for the Weibull accelerated failure time (AFT) model in Appendix B. The critical requirement of our proposed methods to follow is that the survival loss used is differentiable with respect to $W$. For example, the elastic-net-regularized Cox proportional hazards model by Park & Hastie (2007) also satisfies this condition.

## 3 SURVIVAL-SUPERVISED ARCHETYPAL ANALYSIS

Survival supervision can readily be incorporated into the archetypal analysis optimization problem (2.3) by adding a survival loss $L_{\mathsf{surv}}(W, \theta)$ to the objective function, where $\theta$ here is the collection of all parameters specifying the survival model. For example, we could have $L_{\mathsf{surv}}(W, \theta) = L_{\mathsf{Cox}}(\beta|W)$ as defined in equation (2.5) with parameters $\theta = \beta$.[1] Specifically, letting $\Theta$ denote the set of possible values that parameter $\theta$ can take on, and $\eta > 0$ denote a user-specified importance weight of the survival loss, we now instead solve

$$(\widehat{W}, \widehat{H}) \in \underset{\substack{W \in \mathbb{R}^{n \times r}, \, H \in \mathbb{R}^{r \times d}, \, \theta \in \Theta \\ \text{s.t. } W_i \in \Delta^r \text{ for all } i, \, H_g \in \Delta^d \text{ for all } g}}{\operatorname{argmin}} L_{\mathsf{arch}}(W, H; \lambda) + \eta L_{\mathsf{surv}}(W, \theta), \qquad (3.1)$$

where $L_{\mathsf{arch}}$ is given in equation (2.2). Javadi & Montanari (2019) solve the unsupervised archetypal analysis optimization problem (2.3) using the Proximal Alternating Linearized Minimization (PALM) algorithm by Bolte et al. (2014). We augment this algorithm to handle survival supervision, resulting in an algorithm we call SURVIVAL-ARCHETYPES. We first state what SURVIVAL-ARCHETYPES is before explaining our algorithmic modifications to the unsupervised variant.

### 3.1 THE SURVIVAL-ARCHETYPES ALGORITHM

In what follows, we let $\Pi_U(V)$ denote the version of $V$ where each of its rows has been projected onto the set $U$. Formally, the $i$-th row of $\Pi_U(V)$ is given by

$$[\Pi_U(V)]_i = \min_{u \in U} \|u - V_i\|_2.$$

For example, if $V \in \mathbb{R}^{n \times r}$ consists of nonnegative entries where each row's sum is strictly greater than 0, then $[\Pi_{\Delta^r}(V)]_i = V_i / \sum_{j=1}^r V_{i,j}$. Next, we denote the convex hull of the rows of a matrix $V \in \mathbb{R}^{m \times d}$ by

$$\operatorname{conv}(V) := \Big\{ \sum_{i=1}^m w_i V_i : w \in \Delta^m \Big\}.$$

Then the SURVIVAL-ARCHETYPES algorithm repeats the following steps until convergence:

1. Update archetypes: with step size parameter $\gamma_1 = 2\|W^\top W\|_F$, where $\|\cdot\|_F$ denotes the Frobenius norm, set

$$\widetilde{H} \leftarrow H - \frac{1}{\gamma_1} W^\top (WH - \overline{X}),$$

$$H \leftarrow \Pi_{\Delta^d} \Big( \widetilde{H} - \frac{\lambda}{\lambda + \gamma_1} (\widetilde{H} - \Pi_{\operatorname{conv}(\overline{X})}(\widetilde{H})) \Big).$$

2. Update convex combination weights: with step size parameter $\gamma_2$ found using backtracking line search (Parikh & Boyd, 2014, Section 4.3), set

$$W \leftarrow \Pi_{\Delta^r} \Big( W - \frac{1}{\gamma_2} \Big[ (WH - \overline{X})H^\top + \eta \frac{\partial L_{\mathsf{surv}}(W, \theta)}{\partial W} \Big] \Big).$$

3. Update survival model:

$$\theta \leftarrow \operatorname{argmin}_{\widetilde{\theta} \in \Theta} L_{\mathsf{surv}}(W, \widetilde{\theta}).$$

This step amounts to fitting the survival model with rows of $W$ treated as the feature vectors and can just use the model's existing fitting code as a black box.

---

[1]For the Weibull AFT loss given in Appendix B that has parameters $\beta \in \mathbb{R}^r$, $\mu \in \mathbb{R}$, and $\sigma > 0$, we would set $L_{\mathsf{surv}}(W, \theta) = L_{\mathsf{AFT}}(\beta, \mu, \sigma|W)$ as defined in equation (B.1) with parameters $\theta = (\beta, \mu, \sigma)$.

*Initialization.* Following Javadi & Montanari (2019), we use the successive projections algorithm by Araújo et al. (2001) to initialize archetypes $H$. We can initialize each row of $W$ by setting

$$W_i \leftarrow \underset{w \in \Delta^r}{\operatorname{argmin}} \|\overline{X}_i - H^\top w\|_2.$$

Lastly, how the survival model parameters $\theta$ is initialized depends on the survival model. For example, for the Cox proportional hazards model, we can initialize $\beta$ to be the all zeros vector. For the Weibull AFT model, we can initialize $\mu$ and $\beta$ to be all zeros, and $\sigma$ to be 1.

A key technical requirement of SURVIVAL-ARCHETYPES is that we need to be able to compute the gradient $\frac{\partial L_{\text{surv}}(W,\theta)}{\partial W}$. As illustrative examples, we show what this gradient is equal to for the Cox and Weibull AFT models in Appendix C.

## 3.2 RELATING TO THE UNSUPERVISED ARCHETYPAL ANALYSIS PALM ALGORITHM

The original PALM algorithm for unsupervised archetypal analysis can be recovered by setting $\eta = 0$ and removing step 3. Moreover, the step size in step 2 need not be found using backtracking line search. In particular, when $\eta = 0$, step 2 takes a proximal gradient step, where the gradient is

$$(WH - \overline{X})H^\top,$$

which has Lipschitz modulus $2\|HH^\top\|_F$; hence, we can set step size parameter $\gamma_2 = 2\|HH^\top\|_F$ (Bolte et al., 2014, Remark 7(ii)). When $\eta > 0$, the Lipschitz modulus can vary by the survival model used and in general does not have a closed-form expression, so we use a line search. Lastly, if we are not constraining the archetypes to correspond to word distributions, then the projection onto $\Delta^d$ in step 1 can be removed.

## 4 NEURAL SURVIVAL-SUPERVISED TOPIC MODELS

Our proposed approach to a neural survival-supervised topic modeling builds on the SCHOLAR framework by Card et al. (2018). Card et al. do not explicitly consider survival analysis in their setup although they mention that predicting different kinds of real-valued outputs can be incorporated by using different label networks. We use their same setup and have the final label network perform survival analysis via the same approach as Katzman et al. (2018); note that Katzman et al. specifically consider the Cox proportional hazards model but their neural net approach works with some other survival models as well such as the Weibull AFT model. We first give an overview of SCHOLAR and then explain how to implement the final survival analysis label network.

For ease of exposition, we present the SCHOLAR framework without what Card et al. refer to as "covariates" (auxiliary information known about subjects in addition to the word count matrix $X$). The SCHOLAR framework specifies a generative model for the data, including how each individual word in each subject is generated. In particular, recall that $X_{i,u}$ denotes the number of times the word $u \in \{1, 2, \ldots, d\}$ appears for the $i$-th subject. Let $n_i$ denote the number of words for the $i$-th subject, i.e., $n_i = \sum_{u=1}^d X_{i,u}$. We now define the random variable $\psi_{i,\ell} \in \{1, 2, \ldots, d\}$ to be what the $\ell$-th word for the $i$-th subject is (for $i = 1, 2, \ldots, n$ and $\ell = 1, 2, \ldots, n_i$). Then the generative process for SCHOLAR with $r$ topics is as follows, stated for the $i$-th subject:

1. Generate the $i$-th subject's topic distribution:
    (a) Sample $\widetilde{W}_i$ from a logistic normal distribution with mean vector $\boldsymbol{\mu} \in \mathbb{R}^r$ and covariance matrix $\boldsymbol{\Sigma} \in \mathbb{R}^{r \times r}$.
    (b) Set the topic weights vector for the $i$-th subject to be $W_i = \text{softmax}(\widetilde{W}_i)$.
2. Generate the $i$-th subject's words:
    (a) Set word parameter $\eta_i = f_{\text{word}}(W_i)$, where $f_{\text{word}}$ is a generator network.
    (b) For word $\ell = 1, 2, \ldots, n_i$:
        Sample $\psi_{i,\ell} \sim \text{Multinomial}(\text{softmax}(\eta_i))$.
3. Generate the $i$-th subject's output label:
    Sample $Y_i$ from a distribution parameterized by label network $f_{\text{label}}(W_i)$.

There are a wide variety of choices for the parameters $\boldsymbol{\mu}, \boldsymbol{\Sigma}, f_{\mathsf{word}}$, and $f_{\mathsf{label}}$. For example, to approximate supervised LDA (McAuliffe & Blei, 2008) where the topic distributions are sampled from a symmetric Dirichlet distribution with parameter $\alpha > 0$ and the output label is continuous and has unit variance, we set $\boldsymbol{\mu}$ to be the all zeros vector, $\boldsymbol{\Sigma} = \mathrm{diag}((r-1)/(\alpha r))$, $f_{\mathsf{word}}(w) = w^\top H$ where $H \in \mathbb{R}^{r \times d}$ has a Dirichlet prior per row, $f_{\mathsf{label}}(w) = w^\top \phi$ for a parameter vector $\phi \in \mathbb{R}^r$, and set $Y_i$ to be generated from a Gaussian with mean $f_{\mathsf{label}}(w)$ and variance 1. Card et al. (2018) also explain how to approximate the correlated topic model by Lafferty & Blei (2006). To estimate the model parameters, Card et al. use a sampling-based variational autoencoder framework (Kingma & Welling, 2014; Rezende et al., 2014).

*Survival supervision.* To incorporate survival analysis, we follow the same approach as Katzman et al. (2018) and change step 3 of the generative process above to be deterministic and instead output the variable $\Xi_i = f_{\mathsf{label}}(W_i) := \beta^\top W_i$ for parameter vector $\beta \in \mathbb{R}^r$. In particular, we do not actually model how observed times $Y_i$'s are generated; modeling $\Xi_i$'s is sufficient. Then we can minimize the Cox proportional hazards loss:

$$L_{\mathsf{Cox}}(\beta|W) = \sum_{i=1}^n \delta_i \Big[ -\Xi_i + \log \sum_{j=1 \text{ s.t. } Y_j \geq Y_i}^n \exp(\Xi_i) \Big],$$

where $z_i = \frac{\log Y_i - \mu - \Xi_i}{\sigma}$. Regularization on $\beta$ can easily be added (e.g., lasso, elastic net). Other losses are also possible. The Weibull AFT loss given in Appendix B uses the same label network as the Cox example above, namely $f_{\mathsf{label}}(W_i) = \beta^\top W_i$. For both the Cox and Weibull AFT examples, the label network could instead be a multilayer perceptron or a more complex neural net rather than a simple inner product. We refer to SCHOLAR with a survival loss as SURVIVAL-SCHOLAR.

## 5 EXPERIMENTS

We apply SURVIVAL-ARCHETYPES and SURVIVAL-SCHOLAR to two survival analysis datasets focusing on two diseases: pancreatitis and breast cancer. For pancreatitis, we use the MIMIC III electronic health records dataset (Johnson et al., 2016), looking only at the pancreatitis patients admitted to the intensive care unit (ICU) and who did not die while in the ICU; this amounted to 371 patients where we extracted 2557 features (preprocessing details are in Appendix D.1). We predict how long each patient will stay in the ICU. For breast cancer, we use the METABRIC dataset (Curtis et al., 2012), which consists of 1981 patients. We use the same 79 one-hot encoded features as Lee et al. (2018) to predict time until death per subject. Some features are continuous and need to be discretized for use with our topic models (resulting in 100 total features; see Appendix D.2 for details). For both datasets, we randomly divide the dataset into a 75%-25% train-test split.

We benchmark our approaches against a total of 10 baselines: 7 classical methods (lasso-regularized Cox proportional hazards with and without PCA preprocessing, Weibull AFT with and without PCA, $k$-nearest neighbor survival analysis (Beran, 1981; Lowsky et al., 2013) with and without PCA, and random survival forests (Ishwaran et al., 2008)), 2 deep learning methods (DEEPSURV (Katzman et al., 2018) and DEEPHIT (Lee et al., 2018)), and Dawson and Kendziorski's SURVLDA (Dawson & Kendziorski, 2012). For lasso-regularized Cox, our hyperparameter sweep does include an approximation to the standard unregularized Cox model. For simplicity, for our archetypal analysis and neural approaches, we use the standard Cox model as the survival model. For all methods, if the method does not already have a hyperparameter selection procedure, we use 5-fold cross-validation on the training data to select hyperparameters prior to training on the complete training data using the best parameters found; hyperparameter search grids are in Appendix E. For the pancreatitis dataset, due to the number of subjects being small, we use *repeated* 5-fold cross-validation with 10 repeats. Repeated k-fold cross validation has been found to be useful in such small dataset regimes (Braga-Neto & Dougherty, 2004). For both cross-validation and for evaluating test set accuracy, we use the standard survival analysis metric of *concordance index* (Harrell Jr et al., 1982), which is the fraction of pairs of validation/test subjects correctly ordered by the prediction algorithm in terms of which subject has a longer survival time (amongst pairs that can be ordered).

Test set concordances are reported in Table 1. On the pancreatitis dataset, SURVLDA followed by SURVIVAL-ARCHETYPES outperform all the other methods, and the two deep learning baselines (DEEPSURV and DEEPHIT) perform worse than standard Cox proportional hazards as well as many of the other classical baselines. Meanwhile, on the breast cancer dataset, DEEPSURV achieves the best performance although Weibull AFT, $k$-nearest neighbors with PCA preprocessing, DEEPHIT,

| Model | Dataset | |
|---|---|---|
| | Pancreatitis | Breast Cancer |
| Lasso Cox | 0.56 | 0.65 |
| Lasso Cox + PCA | 0.53 | 0.64 |
| Weibull AFT | 0.50 | *0.66* |
| Weibull AFT + PCA | 0.60 | 0.65 |
| $k$-nearest neighbors | 0.59 | 0.58 |
| $k$-nearest neighbors + PCA | 0.53 | *0.66* |
| Random Survival Forest | 0.56 | 0.60 |
| DEEPSURV | 0.55 | **0.67** |
| DEEPHIT | 0.53 | *0.66* |
| SURVLDA | **0.64** | 0.64 |
| SURVIVAL-ARCHETYPES | *0.63* | 0.63 |
| SURVIVAL-SCHOLAR | 0.59 | *0.66* |

Table 1: Test set concordance indices for various methods on the pancreatitis and metabric datasets. Per dataset, we use bold for the highest and italics for the second highest concordance indices.

and SURVIVAL-SCHOLAR all do nearly as well. Also, note that despite SURVIVAL-SCHOLAR being a neural approximation of SURVLDA, the two methods' accuracies are different; this phenomenon has also been reported for SCHOLAR and the various topic models it approximates (Card et al., 2018). Overall, there is no single best survival estimator. The three survival-supervised topic models also jointly estimate topics, and per topic, tells us whether presence of that topic leads to greater or lower probability of survival. As we discuss next, despite SURVIVAL-SCHOLAR having only a concordance index of 0.59 on the pancreatitis dataset, it still manages to produce clinically interpretable topics predictive of whether pancreatitis patients will stay longer in the ICU.

We now give a brief summary of learned topics. Note that for both datasets, the vast majority of words we used require clinical expertise to interpret. For ease of exposition, we defer examples of actual topics learned to Appendix F, where per topic, we list its top 20 most probable words along with the topic's Cox $\beta$ coefficient—a higher coefficient corresponds to predicting a shorter ICU length of stay in the pancreatitis dataset and a shorter time until death in the breast cancer dataset. A topic with $\beta$ coefficient 0 gets ignored for prediction.

*Pancreatitis.* SURVIVAL-ARCHETYPES identified one archetype with a nonzero Cox $\beta$ coefficient (4.5) corresponding to a healthy group with lower-risk interventions (e.g., smaller-bore IV, normal MCV and HCT, top words do not have data elements related to severe illness). All other archetypes have $\beta$ coefficient 0 and correspond to sicker patient characteristics (e.g., atypical lab tests and toxicology panels). SURVIVAL-SCHOLAR separated clinical events into 3 meaningful topics: one for laboratory tests, one for patient presentation characteristics, and one for procedures, precautions, monitoring, and vitals (this last topic has the smallest $\beta$ coefficient associated with longer ICU length of stay). SURVLDA also produced interpretable topics such as critical illness, normal health state, and acid base disorders and liver involvement.

*Breast cancer.* SURVIVAL-SCHOLAR found topics that distinguish elderly, advanced cancers ($\beta$ coefficient 0.71) from ones with early and younger hormone positive characteristics ($\beta$ coefficients $-0.74$ and $-0.79$). SURVLDA also produced topics with identifiable characteristics; however more topics were found (7 topics) with two overlapping topics indicative of elderly stage 2 breast cancer, and three other overlapping topics (all indicative of hormone positive, cellular, and proliferative features). SURVIVAL-ARCHETYPES has noticeably lower prediction accuracy on this dataset, which is reflected in the topics it learns: the two topics with nonzero $\beta$ coefficients have opposite $\beta$ coefficient signs yet have mostly the same top words, suggesting too much topic overlap.

## 6 CONCLUSIONS

Many methodological advances have been made in survival analysis especially with the help of deep learning. The advances have largely focused on prediction accuracy and less on interpreting time-to-event predictions in the application domains of interest. This interpretation can be challenging when the number of features is large and how features relate is not obvious. In this paper, we show that survival-supervised topic modeling can address this challenge: the topics learned reveal feature co-occurrences and have relative weights indicating their impact on predicting longer or shorter survival times. These topics can be used by practitioners to check if the models agree with existing domain knowledge and to help with model debugging. These survival-supervised topic models are flexible and can be used with a variety of topic and survival models.

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

## A ESTIMATING POINT ESTIMATES FOR SURVIVAL TIMES

We now explain how an estimate $\widehat{S}(\cdot|w)$ of the survival function $S(\cdot|w)$ can be used to construct an estimated survival time for feature vector $w$. Recall that random variable $W_0$ denotes a feature vector with survival time $T_0$, another random variable. Note that the survival function $S$ can be written as

$$S(t|w) = 1 - \mathbb{P}(T_0 \leq t \mid W_0 = w),$$

which is 1 minus the CDF of $T_0$ conditioned on $W_0 = w$; we denote this CDF as $F(\cdot|w)$.

Once we have an estimate $\widehat{S}$ for $S$, to predict a single value for the survival time corresponding to test feature vector $w$, the most common approach is to look at where the survival function $\widehat{S}(\cdot|w)$ crosses 1/2 (this is also where CDF $F(\cdot|w)$ crosses 1/2, which is a median of the distribution). Specifically, for feature vector $w$, we find a median survival time estimate $\widehat{T}_0(w)$ to be a time $\tau$ such that $\widehat{S}(\tau|w) \approx 1/2$. For example, we can compute $\widehat{T}_0(w)$ using the equation

$$\widehat{T}_0(w) = \frac{1}{2}\Big[\inf\big\{t : \widehat{S}(t|w) \leq \tfrac{1}{2}\big\} + \sup\big\{t : \widehat{S}(t|w) \geq \tfrac{1}{2}\big\}\Big], \tag{A.1}$$

where in practice the infimum and supremum are taken over the observed times in the training data.

## B   THE SURVIVAL LOSS FUNCTION OF THE WEIBULL ACCELERATED FAILURE TIME (AFT) MODEL

As a second example of a survival model that can be combined with either of our proposed survival-supervised topic modeling approaches, we present the Weibull AFT model.

**Example 2 (Weibull AFT)** *The Weibull AFT model assumes each subject's (possibly unobserved) survival time $T_i$ satisfies*

$$\log T_i = \mu + \beta^\top W + \sigma \varepsilon_i,$$

*where $\mu \in \mathbb{R}$, $\beta \in \mathbb{R}^r$, and $\sigma > 0$ are model parameters, and noise variable $\varepsilon$ has probability density function $f_\varepsilon(z) = \exp(z - \exp(z))$ for $z \in \mathbb{R}$. Under random censoring, maximum likelihood estimation for $\mu$, $\beta$, and $\sigma$ amounts to minimizing the loss*

$$L_{\textsf{AFT}}(\beta, \mu, \sigma | W) := -\sum_{i=1}^{n} \left\{ \delta_i \log f_\varepsilon(z_i) - \delta_i \log \sigma + (1 - \delta_i) \log S_\varepsilon(z_i) \right\}, \tag{B.1}$$

*where*

$$z_i = \frac{\log Y_i - \mu - \beta^\top W_i}{\sigma}, \quad S_\varepsilon(t) = \int_t^\infty f_\varepsilon(u) du = e^{-e^t}.$$

*Note that the AFT model is actually a proportional hazards model with hazard function*

$$h(t|w) = \frac{\exp(-\mu/\sigma) t^{1/\sigma - 1}}{\sigma} \exp(\beta^\top w).$$

*Hence, after minimizing the loss function $L_{\textsf{AFT}}(\beta, \mu, \sigma | W)$ to estimate $\beta$, $\mu$, and $\sigma$, we can plug in their estimates into the hazard formula above and compute an estimate for survival function $S$ using equation (2.6).*

To use the Weibull AFT survival model with our survival-supervised neural topic modeling approach in Section 4, we would minimize loss (B.1), where we redefine

$$z_i = \frac{\log Y_i - \mu - \Xi_i}{\sigma}.$$

As a reminder, $\Xi_i$ here is the output of the label network for subject $i$. In the simplest case, we would set $\Xi_i = \beta^\top W_i$. However, we could replace this inner product with a neural net.

## C   COX AND WEIBULL AFT SURVIVAL LOSS GRADIENTS

For archetypal analysis, Cox and Weibull AFT survival models could both be readily used by using the gradients given below.

**Example 3 (Cox survival loss gradient)** *When $L_{\textsf{surv}}(W, \theta) = L_{\textsf{Cox}}(\beta | W)$ as given in equation (2.5) with $\theta = \beta$, we have*

$$\frac{\partial L_{\textsf{surv}}(W, \theta)}{\partial W_{\ell, g}} = \left( \sum_{i=1 \ s.t. \ Y_i \leq Y_\ell}^{n} \frac{\delta_i \exp(\beta^\top W_\ell)}{\sum_{j=1 \ s.t. \ Y_i \leq Y_j}^{n} \exp(\beta^\top W_j)} - \delta_\ell \right) \beta_g.$$

**Example 4 (Weibull AFT survival loss gradient)** *When $L_{\textsf{surv}}(W, \theta) = L_{\textsf{AFT}}(\beta, \mu, \sigma | W)$ as given in equation (B.1) with $\theta = (\beta, \mu, \sigma)$, we have*

$$\frac{\partial L_{\textsf{surv}}(W, \theta)}{\partial W_{\ell, g}} = -\frac{1}{\sigma} \beta_g \left[ \exp \left( \frac{Y_\ell - \beta^\top W_\ell - \mu}{\sigma} \right) - \delta_\ell \right].$$

## D   DATASET PREPROCESSING

### D.1   PANCREATITIS

The pancreatitis dataset we evaluated on is created from MIMIC III, a critical care health records database containing 52 thousand individuals and their hospital encounters involving admission to

the ICU at Beth Israel Deaconess center between 2001 and 2012 (Johnson et al., 2016). Experiments were conducted using a subset of MIMIC III version 1.4 dataset consisting of patients having pancreatitis requiring admission to the ICU. Patients were included in the study if they have an ICU admission with a primary billing code of pancreatitis, resulting in a cohort of 371 individuals. For patients who are admitted to the ICU multiple times, we only consider their first visit to the ICU. This subset of the data has no right-censoring.

Features extracted include demographics, medications, billing codes, procedures, laboratory measurements, events recorded into charts, and vitals. Features were extracted from the relational database into a 4-column format for *patient id*, *time*, *event*, and *event value*. To prevent erroneous merging of different events into a single event, and to provide more informative events, event strings are concatenations of the event descriptor prefixed with the table from which they are derived and additional relevant information such as measurement type, measurement units, etc. Because events recorded in charts are sometimes automated and sometimes manually entered, a physician-developed mapping and lower-casing all fields were used to resolve duplicate entries. As we aim to predict the patient length of stay in ICU, we extract clinical events from the subjects' electronic health records strictly before ICU admission. Our definitions of clinical events mean that a subject can have multiple instances of one event; for example, one patient might have multiple results for a particular lab test on file.

Single-occurrence categorical events (e.g., gender) are one-hot encoded, with one category removed as the reference category. Multiple-occurrence categorical events (e.g., urine color) are encoded by counting each categorys occurrences in a single subjects records. Numeric clinical events are treated as categorical by mapping observed values to equally spaced ranges by quantile (5 bins of roughly equal number of subjects per bin). Missing records are not imputed as missing certain events can have clinical significance. Therefore, for features with incomplete records, an additional feature is added solely to indicate whether missingness is observed for each subject; this approach to handling missing data is motivated by the work of Lipton et al. (2016). After preprocessing, the total number of features used for prediction is 2557.

### D.2 BREAST CANCER

The Molecular Taxonomy of Breast Cancer International Consortium (METABRIC) dataset contains genetic and clinical information for 1981 subjects with primary breast tumors Curtis et al. (2012). We use the preprocessed version by Lee et al. (2018), who one-hot encoded 21 clinical features to obtain 79 features.[2] Of the 1981 subjects, 55% (1093) are right-censored. After one-hot encoding the categorical features, a total of 79 features are used for prediction in experiments that do not involve topic modeling. As the survival-supervised topic models we use do not handle continuous features, we discretize each continuous feature into 5 equal-sized bins based on feature value quantiles, the same way we did for the pancreatitis dataset. This results in a total of 100 features.

### E   HYPERPARAMETER SEARCH GRIDS

We perform grid search for all models except for DEEPSURV and DEEPHIT, which use their own hyperparameter selection. We specify the search grid per method below.

Lasso-regularized Cox: lasso regularization parameter: $\{0.0001, 0.025, 0.05, 0.075, 0.1, 0.15, 0.25, 0.5\}$

Lasso-regularized Cox + PCA: lasso regularization parameter: $\{0.0001, 0.025, 0.05, 0.075, 0.1, 0.15, 0.25, 0.5\}$; number of PCA components: $\{3, 5, 7, 10, 25\}$

Weibull AFT: no hyperparameter tuned

Weibull AFT + PCA: number of PCA components: $\{3, 5, 7, 10, 25\}$

$k$-nearest neighbor survival analysis: number of nearest neighbors $k$: $\{4, 8, 16, 32, 64, 128, 256, 512\}$

$k$-nearest neighbor survival analysis + PCA: number of nearest neighbors $k$: $\{4, 8, 16, 32, 64, 128, 256, 512\}$; number of PCA components: $\{3, 5, 7, 10, 25\}$

---

[2]https://github.com/chl8856/DeepHit

Random survival forest: number of trees: $\{16, 32, 64, 128, 256, 512\}$

SURVLDA: number of topics $r$: $\{3, 5, 7, 10\}$; Dirichlet $\alpha$: $\{0.1, 10\}$

SURVIVAL-ARCHETYPES: number of archetypes $r$: $\{3, 5, 7, 10\}$; archetypal analysis regularization parameter $\lambda$: $\{10, 100, 1000\}$; survival loss weight $\eta$: $\{10^4, 10^5, 10^6, 10^7\}$

SURVIVAL-SCHOLAR: number of topics: $\{3, 5, 7, 10\}$; survival loss weight $\eta$: $\{10^1, 10^2, 10^3, 10^4, 10^5, 10^6\}$

## F  EXAMPLES OF TOPICS LEARNED

Note that for each survival-supervised topic model learned, what matters in terms of how topics predict survival time is the relative values of the Cox $\beta$ coefficients for that specific model. A $\beta$ coefficient of 0 corresponds to a topic that gets ignored by prediction. Below, we present the topics with nonzero $\beta$ coefficient learned using all training data (with hyperparameters chosen via cross-validation) for SURVIVAL-ARCHETYPES and SURVIVAL-SCHOLAR. Per topic, we indicate the $\beta$ coefficient and list the top 20 most probable words for that topic.

### F.1  PANCREATITIS

SURVIVAL-ARCHETYPES finds 10 topics but only one that has a nonzero Cox $\beta$ coefficient:

| Topic 1 – Cox $\beta$ coefficient 4.54308798 | |
|---|---|
| Word | Probability within topic |
| 20_gauge_insertion_date:::1 | 0.00259127 |
| lab:blood:chemistry:sodium:::~q4(139.0-142.0)_count | 0.00154494 |
| lab:blood:chemistry:urea_nitrogen:::~q1(2.0-12.0)_count | 0.00153877 |
| lab:blood:hematology:mcv:::~q4(90.0-94.0)_count | 0.00152838 |
| lab:blood:hematology:hematocrit:::~q5(36.5-51.8)_count | 0.00147893 |
| lab:blood:chemistry:gamma_glutamyltransferase:::~missing_flag | 0.00147567 |
| lab:blood:chemistry:ntprobnp:::~missing_flag | 0.00147308 |
| lab:blood:chemistry:immunoglobulin_g:::~missing_flag | 0.00147301 |
| chart:cvp_alarm_low:mmhg:::~missing_flag | 0.00147280 |
| chart:cvp_alarm_high:mmhg:::~missing_flag | 0.00147280 |
| chart:alt:iu/l:::~missing_flag | 0.00147145 |
| chart:ast:iu/l:::~missing_flag | 0.00147145 |
| lab:blood:hematology:nucleated_red_cells:::~missing_flag | 0.00147084 |
| lab:blood:hematology:granulocyte_count:::~missing_flag | 0.00147082 |
| lab:blood:chemistry:protein_electrophoresis:::~missing_flag | 0.00147058 |
| lab:blood:chemistry:cholesterol,_ldl,_measured:::~missing_flag | 0.00146990 |
| chart:troponin-t:ng/ml:::~missing_flag | 0.00146951 |
| prescribed:sodium_phosphate_via_iv:mmol:::~missing_flag | 0.00146929 |
| chart:tank_b_psi.:na:::~missing_flag | 0.00146928 |
| chart:tank_a_psi.:na:::~missing_flag | 0.00146928 |

SURVIVAL-SCHOLAR finds three topics, all with nonzero Cox $\beta$ coefficients:

| Topic 1 – Cox $\beta$ coefficient 0.30225980 | |
|---|---|
| Word | Probability within topic |
| lab:blood:chemistry:glucose:::~q2(95.0-112.0)_count | 0.28912985 |
| lab:blood:chemistry:anion_gap:::~q3(13.0-15.0)_count | 0.28761890 |
| lab:blood:chemistry:urea_nitrogen:::~q1(2.0-12.0)_count | 0.28706142 |
| lab:blood:hematology:rdw:::~q4(15.8-17.4)_count | 0.28468764 |
| lab:blood:hematology:hematocrit:::~q3(30.4-33.3)_count | 0.28418380 |
| lab:blood:hematology:platelet_count:::~q2(114.0-184.0)_count | 0.28227943 |
| lab:blood:hematology:mcv:::~q3(87.0-90.0)_count | 0.28181386 |
| lab:blood:chemistry:sodium:::~q3(137.0-139.0)_count | 0.28117300 |
| lab:blood:chemistry:glucose:::~q3(112.0-134.0)_count | 0.28062224 |
| lab:blood:chemistry:potassium:::~q4(4.3-4.8)_count | 0.27537790 |
| lab:blood:chemistry:potassium:::~q3(4.0-4.3)_count | 0.27388600 |
| lab:blood:chemistry:magnesium:::~q4(2.0-2.2)_count | 0.27342620 |
| lab:blood:hematology:mch:::~q4(30.5-31.7)_count | 0.27324143 |
| lab:blood:hematology:rdw:::~q3(14.7-15.8)_count | 0.27201820 |
| lab:blood:hematology:hematocrit:::~q2(27.7-30.4)_count | 0.27129397 |
| lab:blood:chemistry:chloride:::~q3(102.0-105.0)_count | 0.26823607 |
| lab:blood:chemistry:hemoglobin:::~q3(10.2-11.2)_count | 0.26770705 |
| lab:blood:hematology:mch:::~q2(28.1-29.3)_count | 0.26591137 |
| lab:blood:chemistry:urea_nitrogen:::~q5(34.0-128.0)_count | 0.26530087 |
| lab:blood:hematology:red_blood_cells:::~q3(3.4-3.8)_count | 0.26439255 |

| Topic 2 – Cox $\beta$ coefficient $-0.77905321$ | |
| --- | --- |
| Word | Probability within topic |
| chart:admit_wt:kg:::˜missing_flag | 0.37567450 |
| chart:head_of_bed:na:::˜missing_flag | 0.35496010 |
| lab:blood:chemistry:digoxin:::˜missing_flag | 0.35346085 |
| chart:nares_r:na:::˜missing_flag | 0.35066888 |
| prescribed:potassium_phosphate_via_iv:mmol:::˜missing_flag | 0.35065705 |
| chart:radiologic_study:na:::˜missing_flag | 0.34568897 |
| prescribed:atorvastatin_via_po:mg:::˜missing_flag | 0.34543383 |
| prescribed:metronidazole_via_iv:mg:::˜missing_flag | 0.34471878 |
| prescribed:dextrose_5%_via_iv:ml:::˜missing_flag | 0.34357762 |
| prescribed:dextrose_50%_via_iv:g:::˜missing_flag | 0.34312600 |
| chart:cough/deep_breath:na:+/−:::˜missing_flag | 0.34219468 |
| chart:chloride_(serum):meq/l:::˜missing_flag | 0.34123975 |
| chart:pain_management_route/status_#1:na:::˜missing_flag | 0.34082162 |
| chart:inv#2_waveformappear:na:::˜missing_flag | 0.34081900 |
| chart:edema_location:na:::˜missing_flag | 0.34070030 |
| prescribed:heparin_sodium_via_iv:unit:::˜missing_flag | 0.33888385 |
| chart:pain_level_acceptable:na:::˜missing_flag | 0.33872578 |
| chart:paw_high:cmh2o:::˜missing_flag | 0.33695692 |
| chart:resp_alarm_low:bpm:::˜missing_flag | 0.33690770 |
| chart:alt:iu/l:::˜missing_flag | 0.33685216 |

| Topic 3 – Cox $\beta$ coefficient $-1.06347895$ | |
| --- | --- |
| Word | Probability within topic |
| arterial_line_insertion_date:::1 | 0.27548000 |
| chart:resp_alarm_low:bpm:::˜q5(8.0−93.0)_count | 0.26509485 |
| microbiology:blood_culture:::na | 0.25576746 |
| 20_gauge_insertion_date:::1 | 0.25267342 |
| chart:nbp_alarm_high:mmhg:::˜q5(160.0−180.0)_count | 0.24518475 |
| chart:head_of_bed:na:::˜missing_flag | 0.23182970 |
| multi_lumen_insertion_date:::1 | 0.22956896 |
| lab:blood:blood_gas:base_excess:::˜q4(0.0−3.0)_count | 0.22251068 |
| chart:nares_r:na:::˜missing_flag | 0.22050865 |
| lab:blood:chemistry:anti−nuclear_antibody:::˜missing_flag | 0.21709190 |
| chart:nares_l:na:::˜missing_flag | 0.21417704 |
| gu_catheter_insertion_date:::1 | 0.21182594 |
| chart:resp_alarm_−_high:insp/min:::˜missing_flag | 0.21160777 |
| chart:temperature_fahrenheit:?f:::˜missing_flag | 0.20904190 |
| prescribed:vancomycin_via_iv:mg:::˜missing_flag | 0.20473634 |
| lab:blood:chemistry:acetaminophen:::˜missing_flag | 0.20447205 |
| chart:pain_level_acceptable:na:::˜missing_flag | 0.20438956 |
| chart:o2_saturation_pulseoxymetry_alarm_−_low:%:::˜missing_flag | 0.20393686 |
| chart:cough_type:na:::˜missing_flag | 0.20348457 |
| chart:non−invasive_blood_pressure_alarm_−_high:mmhg:::˜missing_flag | 0.20335248 |

## F.2 BREAST CANCER

SURVIVAL-ARCHETYPES finds 3 topics; only 2 have nonzero Cox $\beta$ coefficients:

| Topic 1 – Cox $\beta$ coefficient $102.32197794$ | |
| --- | --- |
| Word | Probability within topic |
| HER2_IHC_status.1 | 0.04181057 |
| Her2_Expr.1 | 0.03842817 |
| histological.1 | 0.03764347 |
| inf_men_status | 0.03623912 |
| ER_IHC_status.1 | 0.03445554 |
| ER_Expr | 0.03336668 |
| Genefu | 0.03303586 |
| HER2_SNP6_state.2 | 0.03154653 |
| int_clust_memb.3 | 0.02787247 |
| grade.2 | 0.02684169 |
| cellularity | 0.02388820 |
| PR_Expz.1 | 0.02362706 |
| PR_Expz | 0.02182748 |
| lymph_nodes_positive:0.0−1.0 | 0.02049815 |
| site.2 | 0.02015589 |
| cellularity.2 | 0.01693353 |
| grade.1 | 0.01609865 |
| group | 0.01588346 |
| Pam50_Subtype.2 | 0.01506807 |
| stage:2.0−4.0 | 0.01466086 |

| Topic 3 – Cox $\beta$ coefficient $-44.67700060$ | |
| --- | --- |
| Word | Probability within topic |
| HER2_IHC_status.1 | 0.04247266 |
| Her2_Expr.1 | 0.04042959 |
| ER_IHC_status.1 | 0.03680399 |
| ER_Expr | 0.03587382 |
| histological.1 | 0.03482885 |
| HER2_SNP6_state.2 | 0.03436751 |
| inf_men_status | 0.03308938 |
| Genefu | 0.03169076 |
| int_clust_memb.3 | 0.02775788 |
| lymph_nodes_positive:0.0-1.0 | 0.02760196 |
| PR_Expz | 0.02661263 |
| cellularity | 0.02389579 |
| grade.2 | 0.02212554 |
| Pam50_Subtype.2 | 0.01950571 |
| group | 0.01930659 |
| PR_Expz.1 | 0.01884195 |
| grade.1 | 0.01851070 |
| cellularity.2 | 0.01631843 |
| Treatment.5 | 0.01531958 |
| stage:2.0-4.0 | 0.01528393 |

SURVIVAL-SCHOLAR finds 5 topics all with nonzero Cox $\beta$ coefficients:

| Topic 1 – Cox $\beta$ coefficient $0.71125180$ | |
| --- | --- |
| Word | Probability within topic |
| NPI:5.05-6.3 | 0.29416186 |
| grade.2 | 0.26243737 |
| Treatment.3 | 0.23528770 |
| PR_Expz.1 | 0.22858755 |
| ER_IHC_status | 0.22115701 |
| age_at_diagnosis:72.44-92.14 | 0.19971548 |
| group.2 | 0.18821794 |
| ER_Expr.1 | 0.18029946 |
| group.1 | 0.16401250 |
| lymph_nodes_positive:3.0-45.0 | 0.14714028 |
| histological.1 | 0.14465626 |
| int_clust_memb.4 | 0.13438721 |
| Her2_Expr | 0.12702571 |
| size:32.0-182.0 | 0.12109776 |
| site.4 | 0.11865168 |
| size:25.0-32.0 | 0.10704651 |
| Genefu.3 | 0.10639679 |
| Pam50_Subtype | 0.10231068 |
| stage:1.1903520209-2.0 | 0.09872285 |
| NPI:4.05-5.05 | 0.09574302 |

| Topic 2 – Cox $\beta$ coefficient $-0.21598756$ | |
| --- | --- |
| Word | Probability within topic |
| stage:0.0-1.0 | 0.40788037 |
| Treatment | 0.31841293 |
| histological.1 | 0.31109884 |
| HER2_SNP6_state.1 | 0.28868553 |
| int_clust_memb.3 | 0.28224890 |
| PR_Expz | 0.26736894 |
| stage:1.1903520209-2.0 | 0.25793970 |
| Treatment.4 | 0.25260258 |
| PR_Expz.1 | 0.25148767 |
| histological.7 | 0.24185723 |
| Her2_Expr.1 | 0.23500897 |
| int_clust_memb.6 | 0.22067617 |
| ER_Expr.1 | 0.21519190 |
| Genefu | 0.21518746 |
| group.1 | 0.21012034 |
| grade.2 | 0.21007878 |
| site.2 | 0.20887140 |
| age_at_diagnosis:72.44-92.14 | 0.20778224 |
| HER2_IHC_status | 0.20411228 |
| size:32.0-182.0 | 0.20289177 |

| Topic 3 – Cox $\beta$ coefficient $-0.65637207$ | |
| --- | --- |
| Word | Probability within topic |
| `HER2_IHC_status.1` | 0.37488750 |
| `HER2_SNP6_state.2` | 0.30100745 |
| `int_clust_memb.3` | 0.25848496 |
| `HER2_SNP6_state.1` | 0.24172206 |
| `Treatment` | 0.22806108 |
| `ER_Expr` | 0.22049715 |
| `inf_men_status.1` | 0.21472536 |
| `lymph_nodes_positive:0.0-1.0` | 0.21002777 |
| `grade.1` | 0.17921847 |
| `histological.7` | 0.17481270 |
| `NPI:3.04-4.03` | 0.17427845 |
| `histological.3` | 0.16505045 |
| `PR_Expz` | 0.16391344 |
| `inf_men_status` | 0.15946893 |
| `Her2_Expr.1` | 0.15669954 |
| `group` | 0.15509687 |
| `int_clust_memb.5` | 0.15460853 |
| `grade` | 0.15454559 |
| `cellularity.1` | 0.15221779 |
| `site.2` | 0.15143663 |

| Topic 4 – Cox $\beta$ coefficient $-0.74412441$ | |
| --- | --- |
| Word | Probability within topic |
| `Her2_Expr.1` | 0.24829830 |
| `lymph_nodes_positive:0.0-1.0` | 0.24766846 |
| `HER2_SNP6_state.2` | 0.21669297 |
| `cellularity` | 0.20731404 |
| `histological.5` | 0.19091570 |
| `histological.1` | 0.18400107 |
| `ER_IHC_status.1` | 0.18253753 |
| `Treatment.3` | 0.17659634 |
| `Pam50_Subtype` | 0.17532045 |
| `PR_Expz.1` | 0.16891813 |
| `Genefu` | 0.15879555 |
| `ER_Expr` | 0.15030436 |
| `HER2_IHC_status.1` | 0.12956520 |
| `histological.7` | 0.12452137 |
| `HER2_IHC_status.2` | 0.12290693 |
| `size:0.0-15.0` | 0.12223840 |
| `site.2` | 0.12201823 |
| `int_clust_memb.3` | 0.11608906 |
| `site.1` | 0.11278336 |
| `NPI:3.04-4.03` | 0.10929855 |

| Topic 5 – Cox $\beta$ coefficient $-0.78542084$ | |
| --- | --- |
| Word | Probability within topic |
| `PR_Expz` | 0.35541302 |
| `Her2_Expr.1` | 0.31779182 |
| `HER2_IHC_status.1` | 0.30835040 |
| `ER_Expr` | 0.30589350 |
| `lymph_nodes_positive:0.0-1.0` | 0.29850137 |
| `HER2_SNP6_state.2` | 0.27557123 |
| `grade.1` | 0.26845238 |
| `group` | 0.26098454 |
| `age_at_diagnosis:48.07-57.87` | 0.24533404 |
| `grade` | 0.24168900 |
| `ER_IHC_status.1` | 0.22550695 |
| `stage:1.0-1.1903520209` | 0.21476460 |
| `NPI:1.0-3.04` | 0.21265814 |
| `NPI:4.03-4.05` | 0.20501652 |
| `site` | 0.19819605 |
| `histological.5` | 0.19405621 |
| `Pam50_Subtype.2` | 0.18937051 |
| `group.3` | 0.16390443 |
| `histological.9` | 0.14986861 |
| `Genefu.1` | 0.14619182 |

