# OpenReview forum: "Topic Models with Survival Supervision: Archetypal Analysis and Neural Approaches"
_ICLR.cc/2020/Conference — Reject_

### Official Review · AnonReviewer1 · 2019-10-20
**Official Blind Review #1**

**Rating:** 3

**Review:**

The paper addresses the problem of survival analysis (predicting time until, e.g., death) using topic modeling. The point of introducing topic modeling here is to gain better insight into what helps predict survival times of unseen test subjects. This contrasts with much of the earlier work in survival analysis, which is mainly or only concerned with getting the best prediction without concern with the “thematic structure” of features. Such a thematic structure could be useful to clinicians, but analysis of the learned topic structure by clinical experts is apparently left for future work. Empirical results on pancreatitis and metabric datasets show that the methods of the paper are comparable to several established baselines.

I think the goals of the paper are well motivated, as time-to-death is not the only measure of interest and clinicians and other domain experts may want to get better insight into why a given model makes certain predictions. The two approaches presented in the paper are mostly on par with (but do not outperform) the best baselines considered in the paper, which would be fine in the approaches were particularly novel and/or the learned “thematic structures” would have demonstrated uses. Unfortunately, I feel the latter conditions are not met, as I have the following concerns:

(1) I find the paper to be relatively incremental in terms of ML methods, and I consider it to be more of an applied ML paper. The paper builds largely upon survLDA (Dawson and Kendziorsky, 2012), using Car et al. (2018)’s way of approximating LDA in a neural framework. Much of the exposition of Survival-Scholar (i.e., section 4) is based on Scholar (Card et al.). The paper presents another method based on archetypal analysis, which adds a supervised term to the loss. The archetypal approach of the paper merely consists of an “algorithmic modification” and otherwise borrows extensively from Javadi and Montanari (2019) and Bolte et al. (2014). In both cases (survLDA extension and archetypal analysis), most of the equations/algorithms can be traced back to these previous papers.

(2) The main point of the paper is to go beyond simply predicting time-to-event outcomes, and instead gaining some insight into the features helping predict time of death. The authors rightly “note that for both datasets, the vast majority of words we used require clinical expertise to interpret”, but it doesn’t seem they have given the learned topics to clinicians to judge whether they make sense or are helpful in any ways. Instead, appendix F provides lots of tables that are difficult to interpret for non-domain experts with little medical background.

(3) I have some more minor concerns (or questions) about the experimental results. See detailed comments.

Overall, I found the paper to be clear and well written, but algorithmic and empirical contributions are I think rather small (small or no empirical gains, incremental ideas).

Detailed comments/questions:

- Table 1: I don’t think it makes sense to label a system as either first or second. With differences of less than 1% and the small size the test set, most of the top systems are probably within the same confidence intervals, which I think the authors should compute and add to the tables. Many of the earlier works cited in the paper (e.g., DeepSurv, DeepHit) provide such confidence intervals, which are crucial in the case of the submission as its dataset is relatively small compared to e.g. SEER used with DeepHit.

- “For all methods, if the method does not already have a hyperparameter selection procedure, we use 5-fold cross-validation on the training data to select hyperparameters prior [..]”. Shouldn’t cross-validation for hyperparameter selection be used with all the methods of the paper, as the paper doesn’t define a validation set? I’m a bit concerned results may not be fully comparable if the selection procedure is not consistent across algorithms (e.g., 5-fold vs other methods). I presume “already have a hyperparameter selection procedure” refers to a procedure applied to the MIMIC dataset (as original hyperparameters borrowed from previous work might not work well). If so, I think the paper should say this explicitly.

- Experimental results consist of single table that leaves much to be desired in terms of understanding why a method works well or not. For example, what about indicating the number of model parameters (for the parametric methods) and learned hyperparameters. With such a small dataset, cross-validation could still be noisy and prone to selecting a bad set of hyperparameters (e.g., significant differences in terms of number of parameters). An indication of how sensitive to hyperparameters the methods are would be useful too.


**Experience Assessment:**

I have published one or two papers in this area.

**Review Assessment: Checking Correctness Of Derivations And Theory:**

I assessed the sensibility of the derivations and theory.

**Review Assessment: Checking Correctness Of Experiments:**

I carefully checked the experiments.

**Review Assessment: Thoroughness In Paper Reading:**

I read the paper at least twice and used my best judgement in assessing the paper.

---

> ### Author Response · Authors · 2019-11-14
> **Response to AnonReviewer1**
>
> Thanks for the very detailed review!
>
> In terms of methods development, we agree that what we are proposing is not a major advance; we are just showing how the approach by Dawson and Kendziorski (2012) can easily be incorporated to more recently developed topic modeling frameworks, resulting in a fairly wide class of survival-supervised topic models. Our paper largely aims to show how well such survival-supervised topic models works in practice. These models can be competitive with some of the best existing baselines while providing additional thematic information via topics. As we stated in our response to AnonReviewer2, among methods that work more or less equally well, we suspect practitioners would favor more interpretable models. To this end, we agree that we can do more to justify why the topics learned are more "interpretable" than, say, looking at features selected via lasso-regularized Cox proportional hazards.
>
> In a future draft of our paper, we will incorporate all of your suggestions:
> - We will no longer label a method as first or second. Following AnonReviewer2's suggestion, we will instead provide statistical significance results. In particular, we will also run every experiment multiple times with different train/test splits to gauge variability in prediction results as well as variability in what sorts of topics are learned.
> - We will unify hyperparameter selection across all methods.
> - We will provide a summary table for the number of model parameters and hyperparameters for the various models (note that we currently do list the hyperparameters in Appendix E).
> - We will discuss how sensitive hyperparameter choices are for different datasets. We agree that the pancreatitis dataset is really small (which is why we used repeated 5-fold cross-validation with 10 repeats). As stated in our response to AnonReviewer2, we will add more datasets including larger datasets.
>
> While in Section 5 of the paper, the topics have been interpreted by a clinician, in the appendix, we indeed did not give a more detailed clinical reading of the topics. We will provide a more detailed qualitative analysis for the tables in the appendix as well.

---

### Official Review · AnonReviewer2 · 2019-10-23
**Official Blind Review #2**

**Rating:** 3

**Review:**

This paper explores the problem of simultaneous learning for topic modeling and survival prediction.  Their contributions are: (1) integrating two different topic modeling approaches with survival models for joint learning and (2) showing results on two medical datasets with some brief analysis of what topics are recovered.  This is an interesting task to be using with topic modeling.  I appreciate that the approach could be used in medical systems where interpretability of models is very important.

I think the high-level idea of the algorithms is fine, but the main drawback is that the experimental section needs to be expanded on, perhaps with larger datasets and more analysis.  Unless more experiments are provided, I might lean towards rejection.  I think it's ok that some of the empirical results are inconclusive, but it should lead to more quantitative error analysis.  I definitely think the smaller size of the datasets could be one factor in the negative results, which is why I think the authors should try experimenting with a larger dataset.

More specific comments and suggestions:
- Each dataset is small (371 and 1981 data points respectively).  This may be too small to effectively train some of these neural models, which I believe may be one reason why they underperform, especially on the pancreatitis data.  In order for the experiments to be more conclusive, maybe you can switch to a different dataset.
- Related to the size of the data, I suspect that many of the differences in the table are statistically insignificant.  Can the authors please specify which of the results are significantly better than the others?
- While I liked the examples in Section F of the appendix, they are difficult to compare between models, and not all of the models are included.  More quantitative analysis is needed to really understand how the models’ learned topics differ in terms of coherency, cohesiveness, interpretability, etc.  Providing really detailed quantitative comparisons might strengthen the analysis section of the paper.
- I liked the preciseness of the descriptions of different algorithms in the background section.  They were all explained nicely and easy to understand.  However, the background section is a bit too detailed at times.  Authors may consider condensing a bit more.

**Experience Assessment:**

I have read many papers in this area.

**Review Assessment: Checking Correctness Of Derivations And Theory:**

I assessed the sensibility of the derivations and theory.

**Review Assessment: Checking Correctness Of Experiments:**

I carefully checked the experiments.

**Review Assessment: Thoroughness In Paper Reading:**

I read the paper at least twice and used my best judgement in assessing the paper.

---

> ### Author Response · Authors · 2019-11-14
> **Response to AnonReviewer2**
>
> Thanks for the comments and suggestions!
>
> In a future draft of this paper, we will incorporate the bulk of your suggestions:
> - We will include more datasets, including ones larger than what we currently consider.
> - We will include statistical significance results. We do indeed suspect that our proposed approaches as well as a number of the best performing baselines do not yield dramatically different predictions and thus the differences in the test set c-index values are not statistically significant. In such a scenario, when a bunch of algorithms are more or less equally good, we suspect practitioners would favor more interpretable models.
> - We will provide more quantitative and qualitative analysis of the topics learned (to gauge topic coherence, interpretability, etc), and also discuss how these compare to the features selected via lasso-regularized Cox proportional hazards.
> - To the extent possible, we will try to condense the background material.

---

### Official Review · AnonReviewer3 · 2019-10-24
**Official Blind Review #3**

**Rating:** 3

**Review:**

The paper considers the problem of interpreting the predictions for survival analysis using topic models. The classical survival analysis problem assumes each datapoint is a subject (X,Y,\delta) where X is a feature vector and Y is a life time or a censoring time depending on whether the subject is dead (when delta=1) or alive (when delta=0). The usual objective here is to predict survival times. The authors assume the features in X are interpretable readings (eg "low blood pressure") and indicate the number of times that reading was observed. Under this setting, such features can also be seen as words with datapoints being (BoW) documents. The goal then becomes that of finding topics that help predict life times on unseen subjects.

My main concerns with this paper are the following:
- setting is very niche and might not be a great fit for ICLR.
- novelty is limited since most pieces were already present in previous work.
- experimental section only uses two datasets with little improvements.  Such small performance gap hardly convinces that the improvement by the proposed method is statistically meaningful.



**Experience Assessment:**

I do not know much about this area.

**Review Assessment: Checking Correctness Of Derivations And Theory:**

I did not assess the derivations or theory.

**Review Assessment: Checking Correctness Of Experiments:**

I assessed the sensibility of the experiments.

**Review Assessment: Thoroughness In Paper Reading:**

I made a quick assessment of this paper.

---

> ### Author Response · Authors · 2019-11-14
> **Response to AnonReviewer3**
>
> Thanks for the feedback. Please see our responses to the other reviewers.

---

### Decision · Program_Chairs · 2019-12-19

**Decision:**

Reject

**Comment:**

The paper proposes two approaches to topic modeling supervised by survival analysis. The reviewers find some problems in novelty,  algorithm and experiments, which is not ready for publish.